# Coupling Up: A Dynamic Investigation of Romantic Partners’ Neurobiological States During Nonverbal Connection

**DOI:** 10.3390/bs14121133

**Published:** 2024-11-26

**Authors:** Cailee M. Nelson, Christian O’Reilly, Mengya Xia, Caitlin M. Hudac

**Affiliations:** 1Department of Psychology, University of South Carolina, 1512 Pendleton Street, Columbia, SC 29208, USA; 2Carolina Autism and Neurodevelopment Research Center, University of South Carolina, 1800 Gervais Street, Columbia, SC 29201, USA; christian.oreilly@sc.edu; 3Institute for Mind and Brain, University of South Carolina, 1800 Gervais Street, Columbia, SC 29201, USA; 4Artificial Intelligence Institute, University of South Carolina, 1112 Greene Street, Columbia, SC 29208, USA; 5Department of Computer Science and Engineering, University of South Carolina, 1244 Blossom Street, Columbia, SC 29208, USA; 6T. Denny Sanford School of Social and Family Dynamics, Arizona State University, Wilson Hall, Floor 3, Tempe, AZ 85287, USA; mengya.xia@asu.edu

**Keywords:** romantic love, electroencephalography (EEG), hyperscanning neuroscience, frontal alpha asymmetry, nonverbal connection

## Abstract

Nonverbal connection is an important aspect of everyday communication. For romantic partners, nonverbal connection is essential for establishing and maintaining feelings of closeness. EEG hyperscanning offers a unique opportunity to examine the link between nonverbal connection and neural synchrony among romantic partners. This current study used an EEG hyperscanning paradigm to collect frontal alpha asymmetry (FAA) signatures from 30 participants (15 romantic dyads) engaged in five different types of nonverbal connection that varied based on physical touch and visual contact. The results suggest that there was a lack of FAA while romantic partners were embracing and positive FAA (i.e., indicating approach) while they were holding hands, looking at each other, or doing both. Additionally, partners’ FAA synchrony was greatest at a four second lag while they were holding hands and looking at each other. Finally, there was a significant association between partners’ weekly negative feelings and FAA such that as they felt more negative their FAA became more positive. Taken together, this study further supports the idea that fleeting moments of interpersonal touch and gaze are important for the biological mechanisms that may underlie affiliative pair bonding in romantic relationships.

## 1. Introduction

Every day, romantic partners use communication to initiate or maintain closeness. Nonverbal connections can be initiated through actions, such as interpersonal touch, postural changes, and shared gaze, and these actions serve as an important aspect of communication that aids in bonding between romantic partners. In fact, more nonverbal connection between romantic partners is associated with greater intimacy [1,2], sexual interest and arousal [3,4], and relationship satisfaction [5,6]. Even though these moments occur rapidly and may be fleeting, nonverbal connections and cues (or a relative lack of connection) can provide evidence of a partner’s romantic feelings and have lasting effects [7,8]. For instance, memories of simple nonverbal cues, such as eye gaze, facial expression, and touch, can serve as important “turning points” in relationships [9].

Despite the short-term and long-term socioemotional and relational implications of nonverbal communication on romantic relationships [6,10], it is unclear how nonverbal connections (even brief moments) may relate to current experiences and biological mechanisms of positive feeling or love. Longstanding evidence from animals and humans suggests that loving attachments are subserved by instantaneous, biobehavioral synchrony wherein biological signals co-relate [11]. To that extent, synchronized physiology may subserve increased feelings of closeness between romantic partners engaged in nonverbal connection. Indeed, romantic partners sharing interpersonal touch demonstrate increased synchronization of electrodermal [12], heart rate [13], and brain activity [14,15]. However, interpersonal touch is only one type of nonverbal connection important to romantic relationships. Shared gaze and close proximity are other examples of nonverbal connection that occur more often in romantic relationships compared to other social partnerships (e.g., friendships) [16]. Yet it is currently unclear whether there are differences in physiological synchrony among romantic partners across varied nonverbal connections. Examining these relationships is important for understanding how brief moments of nonverbal connection aid in developing and maintaining romantic feelings, building successful romantic relationships, and bolstering an individual’s wellbeing. In fact, synchrony is thought to be greatest among romantic partners compared to other relationships [17,18,19,20] and could be responsible for influencing feelings of intimacy and social connection occurring in moments of nonverbal connection that aid in romantic relationship success [1,2,6]. Therefore, this current study sets out to investigate how gaze, interpersonal touch, and the combination of both between romantic partners may differ in the way they influence neurobiological and psychological states (e.g., approach motivation) relevant to building loving connections.

“Hyperscanning” describes the dual collection of biological processes during dyadic interaction and is often used to measure neural synchrony during dynamic and real-time social interactions [21]. Electroencephalography (EEG) hyperscanning research using phase coupling or power correlational approaches is particularly relevant for evaluating neural synchrony between romantic partners engaged in nonverbal connection. For instance, Kinreich and colleagues [22] found that gamma frequencies (30–90 Hz) occurring in temporoparietal regions of the brain (i.e., regions implicated in social skills) are significantly more correlated in romantic couples compared to strangers. Importantly, they also found that this synchrony was greatest during moments of shared gaze. This suggests that nonverbal connection influences neural synchrony between partners; however, this finding was specific to romantic partners engaged in conversation. Indeed, most hyperscanning research on romantic partners examining neural synchrony uses verbal or non-interactive tasks (i.e., joint video watching). This body of literature indicates that neural synchrony is influenced by creativity [18], honesty [20], interpersonal conflict [19], and relationship quality [17], but there still remains a need to understand how nonverbal connection alone influences neural synchrony between romantic partners. 

### 1.1. Frontal Alpha Asymmetry as a Proxy of Approach Motivation in Romantic Partners

Social approach and its counterpart (social avoidance) describes the neurobiological and psychological instinct to engage (or disengage) with selected people [23]. Considering the nature of affiliative pair bonding [24], neural mechanisms that modulate social approach/avoidance [25] are likely synchronized between romantic partners [11]. One important EEG measurement of social approach is known as frontal alpha asymmetry (FAA), which reflects approach or avoidant motivational states in the brain [26,27]. FAA is calculated by the ratio of natural logarithm of alpha power occurring at the right relative to left frontal scalp electrodes (i.e., FAA = ln(right) − ln(left)) [28]. As brain activity measured via the blood-oxygen-level-dependent (BOLD) signal is negatively correlated with alpha power in many social brain regions (e.g., orbitofrontal cortex, superior frontal gyrus), it is thought that decreases in alpha power indicate increases in hemodynamic activity of the brain [29]. Therefore, approach motivation (i.e., the desire to go toward something) [30,31] is generally linked to more positive FAA scores (i.e., left hemodynamic activity) while avoidance motivation (i.e., the desire to move away from something) is linked to more negative FAA scores (i.e., right hemodynamic activity) [32,33]. It is hypothesized that romantic love would elicit more positive FAA due to romantic partners’ motivation to approach one another [34], yet very little empirical work has examined this explicitly. Evidence of alpha asymmetry is present during passive viewing of affiliative movies [35], across occipital regions during a love induction task [36], and while romantic partners are embracing and kissing [37]. However, there is limited work examining approach/avoidance states and the synchronization of these neurobiological states over time during dyadic nonverbal connection. Furthermore, while FAA may become more positive during certain types of interpersonal touch (e.g., massage therapy, embracing) [37,38], it remains unclear if FAA is present in varied nonverbal connections among romantic partners and whether FAA dynamically shifts or synchronizes between romantic partners during nonverbal connection. 

### 1.2. Current Study Objectives 

This current study examines the biological mechanisms and psychological states underlying nonverbal connections between romantic partners through multiple objectives. First, we aimed to better understand the neural correlates supporting nonverbal connections in dynamic, real-time social interactions between romantic partners. Given that previous research has established that varied types of interpersonal touch and shared gaze are important types of nonverbal connection for romantic relationships [9] and may in-fluence FAA [37], we evaluated FAA across five types of nonverbal connection, varying on elements of interpersonal touch (e.g., holding hands, embracing) and visual contact (e.g., mutual gaze) using a hyperscanning approach. Considering that more positive FAA (i.e., increased left hemodynamic activity) is associated with approach processes [32,33], we hypothesized that participants would demonstrate more positive FAA during in-stances of connection (e.g., mutual gaze and touch) relative to no connection (e.g., no gaze or touch). 

Second, to better characterize synchrony among romantic partners while engaged in nonverbal connection, we analyzed correlations of partners’ FAA across time. As research suggests that neural synchrony is greatest among romantic partners compared to other social relationships [17,18,19,20], we believed FAA synchrony would be present across all levels of nonverbal connection. However, as FAA is largest for romantic partners who are embracing and kissing [37] and neural synchrony among romantic partners might be related to arousal [19] that is likely present during an embrace, we hypothesized that FAA would become most concordant while embracing. More specifically, we predicted that while participants were embracing, they would exhibit more positive patterns of FAA (i.e., more approach) that would match their partner’s FAA patterns more closely than the other conditions.

Finally, there is limited research targeting why dyadic nonverbal connections may vary across people, particularly given the variability in how a person perceives their relationship and ongoing, lived experiences. Yet a better understanding of these underlying biological mechanisms may aid in strengthening the connections and bonds within a couple. For instance, nonverbal behavior may strengthen intimate relationships [7], but it is still unclear how the biological mechanisms that underlie moments of nonverbal connection in romantic relationships are linked to other predictors of relationship outcomes (e.g., relationship duration, feelings of intimacy). To initiate this work, we explored associations between multiple individual factors (age, relationship duration, time spent together, romantic/loving feelings, wellbeing) and subject-level biological correlates (FAA, synchrony with their partner). This aim was largely exploratory; therefore, we did not have specific predictions but expected general correlations between biological mechanisms and individual factors. For instance, it would be reasonable to predict people would exhibit more positive FAA following more positive romantic feelings, and that couples that spend more time together may be more attuned to each other’s nonverbal communication. Given the preliminary nature of this work, we conducted this post hoc analysis on the conditions and moments when FAA and synchrony were the greatest, following the first two analyses.

## 2. Materials and Methods

### 2.1. Participants

Fifteen dyads (*N* = 30; 15 females; see Table 1 for characterization) aged 18–40 years that were in a self-affirmed romantic relationship completed several experiments related to a larger study being conducted at a university in the Southeastern United States. Relationship status was best described as dating exclusively or cohabitating (*n* = 20) and married (*n* = 10); no dyad identified as casual dating (i.e., dating multiple people). On an open-ended question about sexuality, a majority (*n* = 25, 83%) of the participants identified as heterosexual or straight, and five others identified as queer, bisexual, omnisexual, or straight/trans. Most participants identified as White (*n* = 22, 76%) and eight others identified as Asian, Lebanese/Arab, or Black. Two participants were left-handed. The local ethical review board approved this project, and all participants gave written informed consent. 

### 2.2. General Procedures

Participants attended two in-person visits. During the first visit, they completed a battery of surveys to capture basic demographic information and questions about their partner. During the second visit, participants completed brief surveys to report on their current loving feelings towards their partner and then completed a set of EEG hyperscanning experiments, including the Nonverbal Connections Paradigm (described below). During the week between visits, participants completed a set of surveys that addressed constructs of perception of love, loving feelings, negative feelings toward their romantic partner, positive feelings toward their romantic partner, and general wellbeing (full survey items available in Appendix A). We characterize the individual difference factors used in this study in Table 1. Except for time questions that were reported in hours, all items were recoded so that values of 1 equaled “strongly disagree” and values of 10 equaled “strongly agree”. We generated subject-level scores by averaging across items based upon the construct. 

### 2.3. Nonverbal Connections Paradigm

Participants were asked to interact nonverbally with their partner in five specific ways while EEG was recorded. Conditions varied based upon physical touch (no touch, holding hands, standing embrace) and visual contact (no gaze with eyes closed, shared gaze; Figure 1). The researcher instructed the pair about which condition was next and left the room to start EEG recording for 2 min. The order of condition was consistent across dyads—(1) No Connection, (2) Gaze Only, (3) Hands Only, (4) Gaze and Hands, and (5) Embrace. The first two pairs scheduled (Pair 1, 4) completed 1.5 min of each condition before the duration of each condition was extended to 2 min. One pair (Pair 1) did not complete the Embrace condition due to scheduling constraints. All participants complied with the nonverbal instruction. Post-task feedback indicated that most participants felt focused (*M* = 7.54, *SD* = 1.93, range 2–10), on a scale of 1–10, where 10 indicated full focus. Most participants (*n* = 27) identified at least one positive feeling (e.g., love, happiness, contentment, safety, warmth) during the task, but three participants described only negative feelings (e.g., boredom, awkwardness, pressure; Appendix A). Nine participants specifically described thinking about being or feeling “connected”, and five participants described thinking about their mutual physical connection (e.g., smell, breathing rate; Appendix A). 

### 2.4. EEG Acquisition, Processing, and FAA Computation

Continuous EEG was recorded from high-density 128-channel geodesic sensor nets using Net Station 5.3 software integrated with two identical EEG high-impedance 400-series amplifiers (Magstim-EGI, Eugene, OR, USA). During acquisition, EEG signals were referenced to the vertex electrode, analog filtered (0.1 Hz high-pass, 100 Hz elliptical low-pass), amplified, and digitized with a sampling rate of 250 Hz. Standard post-processing procedures included bandpass filtering between 0.1–40 Hz and automated artifact rejection using the clean_rawdata plugin in EEGLAB [39]. In line with Delorme’s [40] examination of preprocessing standards, channels were rejected using a rejection threshold of 0.9. Large artifacts were then removed via spectrum thresholding using the pop_rejcont function of EEGLAB (frequency range: 20–40 Hz, threshold: 10 dB). All removed channels were interpolated, and data were re-referenced to average. 

EEG for the entire duration of the nonverbal connection conditions was epoched with 500 ms windows and decomposed by frequency using the default EEGLAB settings for time-frequency analyses. Relative and absolute alpha (8–12 Hz) power was then averaged across channels into two frontal clusters (left channels = E23, E24, E26, E27, E33 right channels = E2, E3, E122, E123, E124) for each trial. FAA was calculated by subtracting the natural log-transformed alpha of the left cluster from the right [i.e., ln(right) − ln(left)]. Preliminary models indicated that both relative and absolute alpha exhibited significant condition differences between left and right hemispheres (*p* < 0.0001 for both). To be consistent with the literature [28], relative alpha was utilized throughout the remainder of the study (see Figure 1). 

### 2.5. Analytic Plan

All statistical analyses were performed using R (version 4.3.1). Linear mixed-effects models were computed using restricted maximum likelihood with Nelder–Mead optimization via the “lme4” package [41]. Estimated marginal means (EMMs) and 95% confidence intervals (CIs) are reported for post hoc testing with false discovery rate correction [42] applied for multiple comparisons. A pictorial representation of the analytic plan and model equations are available in Figure 2.

First, mixed-linear-effects models tested group-level (i.e., across all individuals) FAA differences related to condition and dynamic shifts over the two-minute experience by including effects of time to assess a linear slope and quadratic slope across epoch. A random intercept for each person was included to account for repeated measures. Missing data (e.g., due to artifacts) was ignored at the epoch level. Each participant contributed at least 38 valid epochs per condition, with most participants contributing over 200 valid epochs per condition (*M* = 218.7, *SE* = 3.1 500 ms epochs). Dynamic shift characterization was confirmed by extracting model-estimated time features (linear slope, quadratic slope) every 60 epochs (30 s increments) and verifying non-zero values (e.g., confidence intervals do not cross zero) for each condition using lstrends() from the “emmeans” package in R [43]. 

Second, we examined within-dyad effects of FAA across condition and temporal lag by converting FAA scores into within-dyad concordance values. To do this, within each person, we averaged FAA values across four contiguous epochs that generated 60 non-overlapping bins (2 s each bin). Epochs included only data surviving artifact detection and correction; thus, any bin-level missing data were due to missing data (e.g., due to artifacts) across the full 2 s bin. Note that there were 45 bins for the two pairs that only completed 1.5 min per condition. These bins were then used to examine concurrent and lagged coupling (i.e., neural synchrony) of FAA within dyads. Then, we examined concurrent and lagged similarities between the two people within the dyad using concordance correlation coefficients (CCCs; scale limits = −1 to 1) [44,45] at each bin. Positive CCC values indicate strong concordance of FAA (e.g., both FAA reflective of approach), and negative CCC values indicate strong discordance of FAA (e.g., FAA reflects approach in one partner and avoidance in the other partner). Close-to-zero CCC values reflect no linear relationship between partners’ FAA. Although the CCC is more commonly used to validate against a gold-standard measure, it has proved useful in prior dyadic EEG and biological measurement studies [46,47]. In addition to concurrent coupling across bins, we examined whether each person’s FAA was predicted by their partner’s previous FAA values at a 1-bin, 2-bin, and 3-bin lag (see Figure 2). CCC was estimated separately for each condition, only including bins where FAA was available for both partners. Average CCC values were extracted for the entire sample first to describe overall trends, such that computation was tested on data from each participant relative to their partner for each lag and condition, across all available epochs. Then, CCC values were extracted to generate each person’s unique CCC value for each lag and condition to be used for statistical models. In this way, concurrent lag was identical for a participant and their partner (i.e., FAA similarity at the same time point); subsequent lags were not identical because one partner may respond to their partner’s nonverbal cues differently or at different rates. These individually derived CCC values were input into linear mixedeffects models to predict CCC differences by included fixed effects of condition, lag, and a random intercept for each person to account for repeated measures. A priori pairwise comparisons were planned to examine condition differences at each lag level. 

Finally, as a post hoc exploratory analysis we used Pearson correlations with FDR-corrections to estimate how subject-level FAA and within-dyad CCCs related to baseline individual differences in age, relationship duration, estimate of weekday and weekend hours spent with their romantic partner, perception of love, and wellbeing. Given that participants responded to individual difference measures for a week, we also evaluated relationships between FAA, CCCs, and weekly mean and variability from evening daily diary reports of perception of love, loving feelings, positive feelings, negative feelings, wellbeing, and amount of time shared together. For exploratory analyses, we opted to retain all data points as a first step to learn about the variability and potential relationships within data [48]; however, model results where outliers were winsorized can be found in Appendix A. 

## 3. Results

### 3.1. General FAA Differences Related to Condition and Time

Results for the linear mixed-effects model are reported in Table 2 and illustrated in Figure 3. 

#### 3.1.1. Lack of FAA for No Connection and Embrace Conditions 

Post hoc testing following a main effect of the condition indicated positive but non-significant FAA values (i.e., neutral neurobiological state) for both the No Connection condition (EMM = 0.120, CI = −0.06–0.3) and the Embrace condition (EMM = 0.116, CI = −0.06–0.3). Both conditions were less positive than the other conditions, *p* < 0.0001, that indicated positive FAA: Gaze Only (EMM = 0.184, CI = 0.002–0.37), Hands Only (EMM = 0.184, CI = 0.002–0.37), and Gaze and Hands (EMM = 0.182, CI = 0.0008–0.36).

#### 3.1.2. Dynamic Shifts Across Epoch Varied by Condition 

The results for the omnibus model are reported in Table 2, and patterns are illustrated in Figure 3B. As described above, statistical verification of dynamic shifts was computed via testing linear trends from estimated marginal means in 60-epoch (30 s) increments. With minor exceptions, FAA was stable for two conditions: No Connection and Embrace conditions (negative slope slowing down due to negative quadratic term at epoch 180 and epoch 60, respectively). The other conditions (Gaze Only, Hands Only, Gaze and Hands) exhibited an initial increase in FAA (positive slope) that decreased by epoch 120 (negative quadratic term).

### 3.2. Interplay of FAA Responses Within Romantic Partner Dyads

Descriptively (group-level; see Figure 4), conditional CCC values ranged from −0.68 to 0.42, negatively skewed for the Gaze Only condition, and kurtosis values greater than 2 indicated a more peaked distribution (i.e., relative to normal) for all conditions except Hands Only. As illustrated in Figure 4, CCCs were weakly negative (i.e., −0.05 to −0.2) at concurrent, Lag 1, and Lag 3. At Lag 2, group-level CCCs were positive for all conditions, indicating that FAA values were concordant with partner’s FAA values at a 4 s lag. 

For subject-level descriptives (i.e., each person in reference to their partner), see Table 3 (lag-level descriptives available in Appendix A). Results for the linear mixed-effects model indicated a main effect of condition, *F*(4, 543) = 4.10, *p* = 0.003, and highlighted several patterns—(1) more negative CCCs for No Connection and Embrace, relative to Gaze and Hands, *p* < 0.043; (2) more negative CCCs for Gaze Only relative to No Connection and Hands Only, *p* < 0.02. Effects of lag, *F*(3, 543) = 2.37, *p* = 0.07, and the condition by lag interaction, *F*(12,543) = 1.44, *p* = 0.14, were not significant; however, planned comparisons of condition across lag indicated the condition differences described were present only for Lag 2, *p* < 0.005, but no other lag, *p* > 0.13.

### 3.3. Influence of Individual Differences

As FAA was approach-like and CCCs at Lag 2 were most concordant during the Gaze and Hands condition, we only report results specific to this condition. Pearson correlations with FDR corrections revealed no significant relationships between FAA and baseline individual differences (see Appendix A). However, there was a positive correlation between FAA and weekly mean negative feelings, *r*(28) = 0.49, *p* = 0.042 (Figure 5). No Pearson correlations were significant with FDR correction for any of the variables and CCCs. Statistical results for weekly and baseline values are reported in Appendix A. 

## 4. Discussion

In this study, we used an EEG hyperscanning task to evaluate how nonverbal connections influence individual and concordant neurobiological states between romantic partners. Specifically, we investigated how neurobiological states of approach/avoidance as reflected by FAA shifted and synchronized between partners across different levels of nonverbal connection (e.g., hand holding, embracing, mutual gaze). While some of the literature has examined either neural synchrony or FAA between romantic partners, to our knowledge, this is the first study to examine FAA synchrony in the context of nonverbal connection. Our results suggest that FAA varies based on the type of nonverbal connection and how long partners are engaged in nonverbal connection. Additionally, we found that FAA becomes synchronized between romantic partners at a four second lag, particularly when holding hands and looking at each other. 

### 4.1. Approach-like Neural Correlates Vary over Different Types of Nonverbal Connection and Time

In general, group-level FAA was positive for all conditions, suggesting that there was some level of approach for romantic partners across all types of nonverbal connection. While it was initially surprising that there was positive FAA (i.e., indicating approach-like signatures) for the No Connection condition, further examination of participants’ thoughts and feelings during the experiment might best explain these results (see Appendix A). For example, after completing this experiment, one participant indicated that during this condition they were trying to listen to their partner’s breath and make their own breathing patterns apparent to their partner. As such, positive FAA in the No Connection condition might have been caused by participants being in close proximity with their partners in a loving context. 

Results indicated significantly less positive group-level FAA (i.e., less approach) in the No Connection and Embrace conditions compared to the other three conditions (i.e., Gaze Only, Hands Only, Gaze and Hands). Additionally, while our analyses did not reveal a significant main effect of time on FAA, there was a significant interaction term between time and condition. This may suggest that FAA changes dynamically over time based on the type of nonverbal connection romantic partners are engaged in. While FAA was quickly extinguished for the No Connection and Embrace conditions, FAA remained steady until about halfway through the experiment for the other conditions, as confirmed with testing the linear trends in 30 s increments. This may suggest that FAA became increasingly more approach-like as participants engaged in shared gaze, interpersonal touch via hand holding, or both. However, time alone was not a significant predictor, and these results should be interpreted with caution.

The lack of positive FAA and early decrease to FAA below statistically significant levels in the Embrace condition was not in line with our hypothesis nor previous research that suggests alpha asymmetry for romantic partners is largest while embracing or kissing [37]. Even though hyperscanning allows for more naturalistic measurement of dynamic social situations, there are still methodological constraints that might explain the lack of approach signatures during the Embrace condition. For example, unlike previous research (e.g., [37]), partners were asked to embrace longer (i.e., two minutes), which may have felt a little unnatural. Likewise, Packheiser and colleagues [37] conducted their study in partners’ homes, which may have allowed for more comfort between partners. While most of our participants (*n* = 27) indicated positive feelings post-experiment, some (*n* = 3) indicated that they felt awkward, bored, and frustrated. Finally, previous research examining alpha asymmetry during embrace has yet to compare this condition to other types of nonverbal connection (e.g., shared gaze, hand holding), which proved in this current study to elicit stronger approach-like signatures. In fact, Packheiser and colleagues [37] had participants look at each other while embracing, which may suggest that gaze could have been influencing the alpha asymmetry in their study. As our findings support previous research that has demonstrated approach signatures during interpersonal touch [15] and shared gaze [22] separately, there is a need for future research to further distinguish the biological mechanisms underlying differing nonverbal connections in ecologically valid ways. 

### 4.2. Partners’ Brains Couple-Up at a Lagged Offset During Nonverbal Connection

Condition effects of FAA concordance between partners revealed larger concordance (i.e., coupling or synchrony) during the Gaze and Hands condition specifically compared to the No Connection and Embrace conditions. While it was surprising that embracing did not elicit the greatest synchronization of FAA between partners, this may have been due to the weak approach-like neurobiological states (i.e., positive FAA) in general during this condition. Stronger concordance during moments of mutual gaze and interpersonal touch, however, is supported by previous research. For example, greater neural synchronization between romantic partners during interpersonal touch compared to vocal communication has been reported in an fNIRS study [15]. Additionally, it has been reported that synchrony between romantic partners having a conversation was greatest during moments of shared gaze [22]. 

The current results also suggest that concordance effects were greatest at a four second lag. In other words, one participant’s current FAA was most in-sync with their partner’s FAA from four seconds ago. While this was not necessarily expected, given that nonverbal connection occurs rapidly, varied lagged inter-brain concordance ranging anywhere from 1.55 [49] to 12.5 s [50] is reported in hyperscanning studies investigating social interactions [51] and can develop as a result of experimental settings or external environmental changes regardless of social interaction [52]. Additionally, memories of eye gaze, facial expression, and touch are particularly salient in romantic relationships [9]. Therefore, the lagged synchrony might be a result of deeper encoding of the nonverbal connection occurring between romantic partners. For instance, if participant A, in a state of approach, tried to make their partner (i.e., participant B) laugh or smile while engaged in nonverbal connection, participant B’s brain might have been delayed in reaching a state of approach due to encoding and information processing of participant A’s behavior. Additionally, one participant reported initially feeling awkward during the different nonverbal connection conditions but “slowly […] felt more admiration and compassion for my partner thereafter”, as they remained engaged in the nonverbal connection (see Appendix A). This may indicate that because partners initially felt uncomfortable during each nonverbal connection, the synchrony of their neurobiological states might have been delayed and dependent on each other, feeling more comfortable as time went on. 

### 4.3. Negative Feelings May Influence Partners’ Approach-like Neural Correlates

As a preliminary analysis, we explored associations between individual differences and romantic partners’ FAA and CCC values, which revealed only one significant positive correlation between weekly mean negative feelings and FAA. This suggests that as partners’ negative feelings increased, neurobiological states became more approach-like. Considering this measure described negative feelings like anger and tension, these results are in line with evidence that suggests certain negative emotions (e.g., anger), elicit approach-like states [53,54,55]. A visual examination of this relationship demonstrates one outlier that had much more positive FAA and more negative feelings than the sample, however. Thus, we caution against overinterpretation of this relationship and encourage deeper future examinations. Additionally, while no other significant relationships survived corrections for multiple comparisons, there was an uncorrected trend between average weekly negative feelings and CCCs, in line with the FAA association. Here, partners that reported more negative feelings had more concordant neurobiological states. These exploratory findings may be in line with previous research suggesting that romantic partners that are distressed or in moments of conflict are more neurally synced [17,19] but also deserve further exploration in the future. 

### 4.4. Limitations and Future Directions

The order of nonverbal connection conditions was fixed across participants, such that No Connection was always presented first, and Embrace was always presented last. This fixed order may have contributed to the lack of positive FAA in the Embrace condition, as participants may have been fatigued by the time they met this condition. Additionally, EEG data are particularly sensitive to body and head movements that occur naturally when people embrace. While we attempted to mitigate this noise during the experiment (i.e., asking participants to hold the same embrace the entire time, reminding participants to move as little as possible) and in preprocessing of the data (e.g., artifact detection/rejection described in Section 2.5), it is possible that noisy data may have influenced our results. Given that other studies have found positive FAA while romantic partners are embracing, future research should aim to investigate randomized or different orders of nonverbal connections to better understand the relationship between FAA and embracing among romantic partners. 

The sample size in this current study is representative of other within-subject studies investigating FAA [37,56,57]; however, a larger sample size may produce larger variability of FAA and CCCs between nonverbal connection conditions and will be better suited to improve our understanding of the relationship between individual differences, FAA, and CCCs associated with nonverbal connection. Similarly, individual differences at the neural level (e.g., skull thickness, baseline alpha power) and demographic level (e.g., race, relationship type) may influence FAA scores [58]. As such, investigating these constructs in a more diverse sample of romantic partners (e.g., non-white, Hispanic/Latine) and romantic partnership types (e.g., non-monogamous, casually dating) while correcting for overall alpha power within an individual could strengthen the generalizability of these conclusions.

While we chose to focus on concordance of partners’ neurobiological states via FAA to clarify how partners were aligned in terms of approach and avoidance during nonverbal connection, continued work utilizing more sophisticated signal-processing techniques is needed. For example, Granger Causality or Partial Directed Coherence may be particularly helpful in understanding the directionality by which information is sent between people [21]. These techniques would offer more sophisticated insight into the biological process, which could aid in a deeper understanding of individual differences that may influence neural synchrony and co-occurring psychological states. One study examining neural synchrony underlying cooperation between romantic partners using Granger Causality analyses found that there was stronger directional synchronization from females to males during a cooperative task [59]. Future research may also consider incorporating other physiological measures to better understand the relationship between biological synchrony and relationship outcomes. For instance, research has demonstrated that synchronous electrodermal activity predicts mutual romantic and sexual interest [60], and sexual satisfaction moderates heart rate synchrony [61]. While understanding causal interactions and how neurobiological synchrony aligns with the synchronization of other physiological rhythms among romantic partners engaged in nonverbal connection was outside the scope of this current study, the presence of lagged synchrony in our results demonstrates a need to better understand this in future research. 

## 5. Conclusions

This study was one of the first to dynamically examine the impact of various nonverbal connections on romantic partners’ neurobiological states. Our results suggest that all types of nonverbal connection elicit approach-like neurobiological states among romantic partners. Importantly, these neurobiological states appear to synchronize between romantic partners specifically when holding hands and looking at each other. Additionally, this synchrony may be delayed as one partner encodes and processes their partner’s behavior and nonverbal cues. Finally, approach-like neurobiological states seem to be positively related to negative feelings between partners. These findings are particularly relevant to understanding the underlying relationships between nonverbal connections and romantic relationship success. More specifically, this current study may inform future studies interested in investigating how neurobiological states of nonverbal connection influence short- and long-term romantic outcomes.

## Figures and Tables

**Figure 1 behavsci-14-01133-f001:**
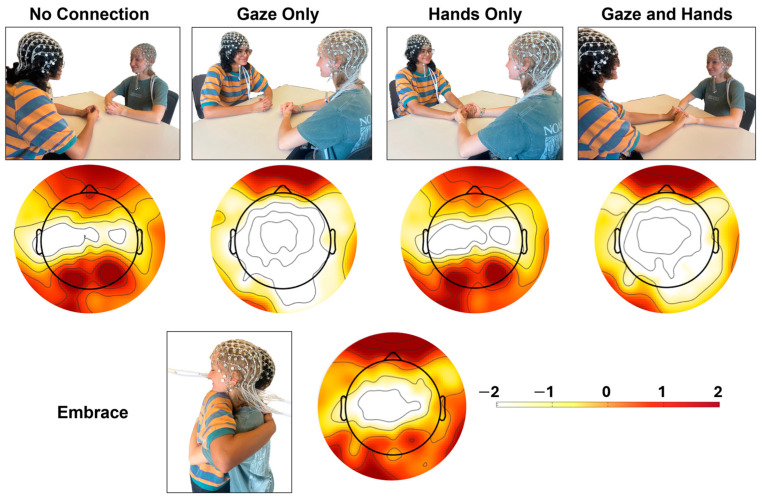
**Nonverbal Connection Conditions and Topographic Maps for Corresponding Conditions.** Topographic maps represent the power spectral density of alpha (8–12 Hz) across all channels for each condition, where darker colors indicate greater power and lighter colors indicate lesser power.

**Figure 2 behavsci-14-01133-f002:**
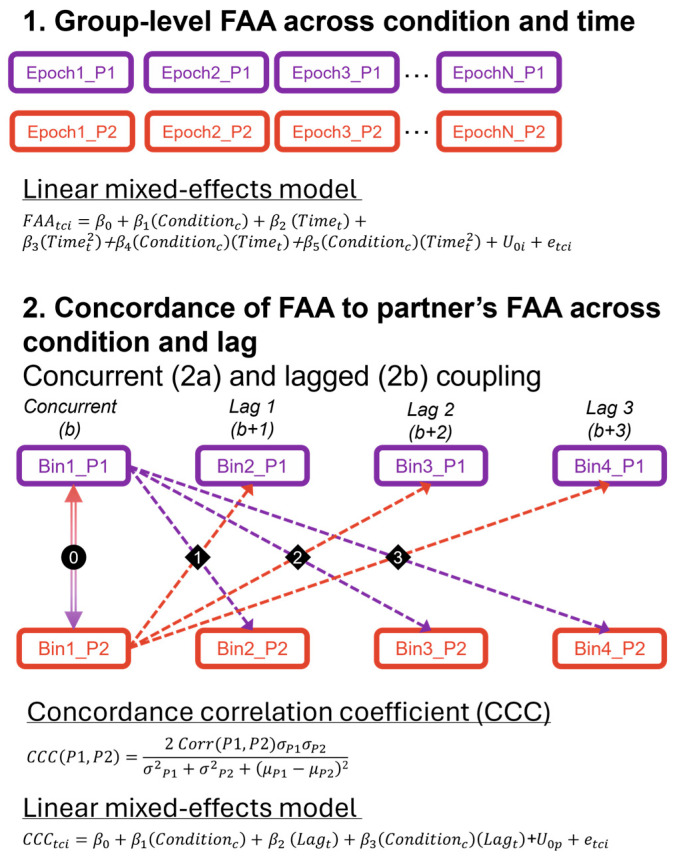
**Analytic Plan.** Graphical representation and model equations for analytic plan to assess (1) group-level frontal alpha asymmetry (FAA) effects across condition and time, and (2) within-dyad FAA effects across condition and temporal lag (1-lag, 2-lag, and 3-lag). Epochs are extracted every 500 ms from continuous EEG. Bins are computed as the average across 4 contiguous epochs (e.g., 2 s). P1 and P2 represent partner one and two, respectively, within a pair.

**Figure 3 behavsci-14-01133-f003:**
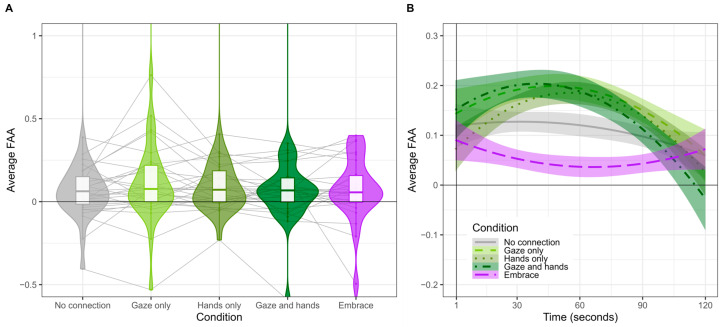
**Average Frontal Alpha Asymmetry (FAA).** Positive FAA values reflect approach processes, zero-like FAA values reflect neutral processes, and negative FAA values reflect avoidance processes. Panel (**A**) illustrates group-level FAA as violin plots to describe variability and boxplots to describe quartile ranges per condition. Gray lines reflect within-person individual differences across condition. Panel (**B**) illustrates group-level average FAA across epoch.

**Figure 4 behavsci-14-01133-f004:**
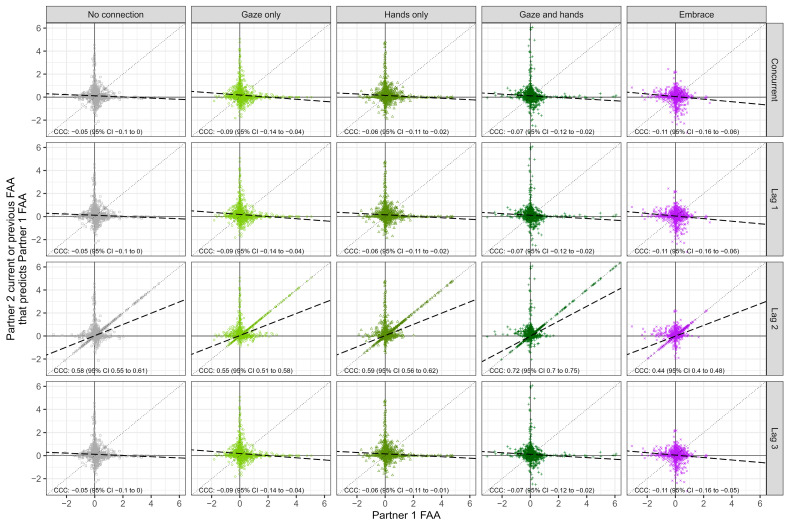
**Evidence of Within-Dyad Concordance Across Condition and Epoch Lag.** Concordance correlation coefficients (CCCs) and 95% confidence intervals are computed and reported. Perfect concordance occurs at CCC = 1 (all points perfectly aligned on the dotted line at 45 degree) and perfect discordance occurs at CCC = −1. Within a dyad, Partner 1 and Partner 2 designations were randomly assigned, as CCCs are symmetric with respect to these attributions (i.e., CCC(P1, P2) = CCC(P2, P1)). Dots illustrate the dyad- and bin-level, such that each dot indicates the concordance between each partner at each bin with valid data. The thick dashed line represents the slope concordance between Partner 1 and 2. Panel rows reflect bin-lag; for example, in row two for Lag 1, Partner 1’s FAA is predicted by Partner 2’s previous bin FAA.

**Figure 5 behavsci-14-01133-f005:**
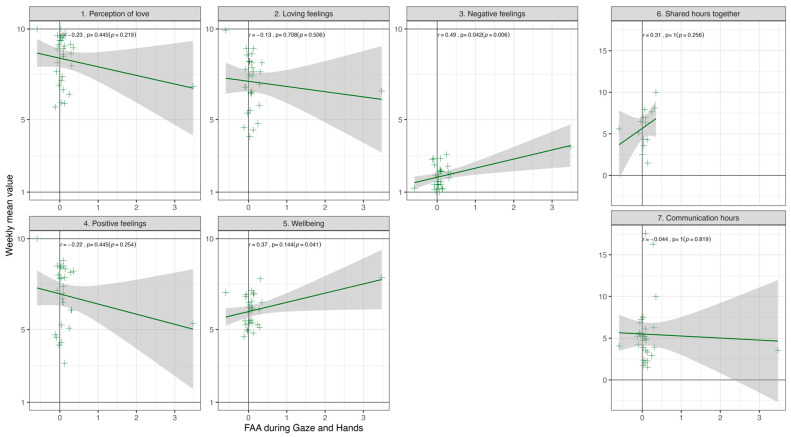
**Relationship Between FAA and Average Weekly Means.** FAA values are extracted as each person’s average FAA value during Gaze and Hand condition, as the condition by which FAA was positive, reflecting more approach-like neurobiological states. Individual difference factors are computed as the weekly means from daily surveys, and specific questions are available in Appendix A. For panels 1–5, items are scored from strongly disagree (1) to strongly agree (10).

**Table 1 behavsci-14-01133-t001:** **Demographic Information for Romantic Partners.** Baseline and weekly averaged descriptive statistics for individual factors are reported for the entire sample. Items used to generate individual factor scores for each construct can be found in Appendix A.

Baseline (Visit 1)	Mean (SD)	Range
Age (Years)	28.03 (5.05)	21–40
Relationship duration (Years)	7.41 (5.99)	1–20
Weekday time (Hours)	40.8 (19.18)	10–90
Weekend time (Hours)	75.9 (19.07)	10–97
Perceived love	9.03 (1.03)	7–10
Wellbeing	6.65 (1.36)	3.59–9.39
**Weekly Mean Across 7 Days**	**Mean (SD)**	**Range**
Perceived love	8.31 (1.33)	5.7–10
Loving feelings	7.07 (1.47)	4.07–9.93
Negative feelings	1.9 (0.67)	1–3.5
Positive feelings	6.86 (1.68)	3.14–10
Wellbeing	6.07 (0.88)	4.61–7.85
Shared time together (Hours/day)	5.83 (2.29)	1.5–10
Communication time (Hours/day)	5.46 (3.63)	1.52–17.57

**Table 2 behavsci-14-01133-t002:** **Linear Mixed-Effects Model Results Predicting Frontal Alpha Asymmetry (FAA).** Model predicting FAA included single-epoch FAA values for each person (*N* = 30), condition (No Connection, Gaze only, Hands only, Gaze and Hands, and Embrace), and epoch (up to 240 epochs).

Effect	*F*	(df1, df2)	*p*-Value
Random intercept	2.99	(1, 32319)	0.0835
Condition	8.27	(4, 32319)	**<0.0001**
Time (slope)	1.62	(1, 32319)	0.2031
Time (quadratic)	1.94	(1, 32319)	0.1639
Condition × Time (slope)	4.09	(4, 32319)	**0.0026**
Condition × Time (quadratic)	6.13	(4, 32319)	**0.0001**

*Note:* Bolded *p*-values indicate significant effects.

**Table 3 behavsci-14-01133-t003:** **Descriptive Statistics of Concordance Correlation Coefficients (CCCs) for Each Condition at the Subject Level.** CCC values were computed for each subject in reference to their partner and are described as collapsed across persons and lag.

	Mean	SD	Min	Max	Skew	Kurtosis
No Connection	0.01	0.102	−0.254	0.389	0.749	2.855
Gaze Only	−0.024	0.109	−0.682	0.228	−1.994	9.742
Hands only	0.011	0.088	−0.213	0.33	0.308	1.415
Gaze and Hands	0.014	0.082	−0.238	0.404	0.792	4.266
Embrace	−0.016	0.102	−0.417	0.293	−0.245	2.642

## Data Availability

The raw data supporting the conclusions of this article will be made available by the authors on request.

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
