# Peer review of "Coupling Up: A Dynamic Investigation of Romantic Partners’ Neurobiological States During Nonverbal Connection"

_behavsci, 2024, doi:10.3390/bs14121133_

Round 1

Reviewer 1 Report

Comments and Suggestions for Authors

The authors use an EEG measurement (FAA) to study the possible neural mechanism of romantic love in nonverbal communication. The paper is well written and clear, the methods and results are described appropriately, and the discussion is consistent with the evidence found. Only minimal typographical errors were found, and minimal suggestions were proposed:

1)     In some parts the full stop is missing at the end of the sentence as: lines 40, 85. Please check for other possible minor typos in the text.

2)     In the correlation analysis, as well described in the methodological section, outliers were included. It would be interesting to show, in the supplementary materials for example, the same analysis without them, as the sample used in main analyses.

3)     The authors did not verify the psychometric characterization of the single participants. Also, if it is already known that such a characterization can modify FAA values ​​mainly under rest conditions, I suggest adding the reference (Smith et al., 2017) to clearly state this issue, or alternatively describing it in the limitations section.

Finally, in the discussion section (line 635) the FAA condition of No Connection is characterized as positive, but in the results as zero-like values. Since the CI values ​​for this condition (as well as for the Embrace condition) are around 0, it might be better to define this value as having a "positive trend".

Biblio:

Smith EE, Reznik SJ, Stewart JL, Allen JJ. Assessing and conceptualizing frontal EEG asymmetry: An updated primer on recording, processing, analyzing, and interpreting frontal alpha asymmetry. Int J Psychophysiol. 2017;111:98-114.

Author Response

Thank you very much for taking the time to review our manuscript. We believe the changes we have made have significantly improved the manuscript. Please find our response to your comments (bold) below in italics. We also have highlighted major changes in the manuscript and have tracked-changes for minor word and grammatical changes. 

The authors use an EEG measurement (FAA) to study the possible neural mechanism of romantic love in nonverbal communication. The paper is well written and clear, the methods and results are described appropriately, and the discussion is consistent with the evidence found. Only minimal typographical errors were found, and minimal suggestions were proposed:

Comment 1: In some parts the full stop is missing at the end of the sentence as: lines 40, 85. Please check for other possible minor typos in the text.

Response 1: Thank you for pointing this out to us. We have corrected all minor typos in the text. Additionally, we have made some other minor wording changes (highlighted by tracked changes) to aid readability.

Comment 2: In the correlation analysis, as well described in the methodological section, outliers were included. It would be interesting to show, in the supplementary materials for example, the same analysis without them, as the sample used in main analyses.

Response 2:  As this analysis was meant to be exploratory, we opted to retain the outliers to better understand true variability in our sample. However, we have conducted an analysis where outliers were winsorized which is now available in supplemental information as Figures S4-7. We have added the following information to section 2.6:

“For exploratory analyses, we opted to retain all data points as a first step to learn about the variability and potential relationships within data [48]; however, model results where outliers were winsorized can be found in supplemental information (Figures S4-7).”

Comment 3: The authors did not verify the psychometric characterization of the single participants. Also, if it is already known that such a characterization can modify FAA values ​​mainly under rest conditions, I suggest adding the reference (Smith et al., 2017) to clearly state this issue, or alternatively describing it in the limitations section.

Response 3: Thank you for pointing us to this reference. We now discuss the need for correcting for individual alpha differences in section 4.4 and reference Smith et al., (2017).

“Similarly, individual differences at the neural level (e.g., skull thickness, baseline alpha power) and demographic level (e.g., race, relationship type) may influence FAA scores [59].  As such investigating these constructs in a more diverse sample of romantic partners (e.g., non-white, Hispanic/Latine) and romantic partnership types (e.g. non-monogamous, casually dating) while correcting for overall alpha power within an individual could strengthen the generalizability of these conclusions.”

Comment 4: Finally, in the discussion section (line 635) the FAA condition of No Connection is characterized as positive, but in the results as zero-like values. Since the CI values ​​for this condition (as well as for the Embrace condition) are around 0, it might be better to define this value as having a "positive trend".

Response 4: We agree that this interpretation makes more sense and is more aligned with how we interpret the results in the discussion. We have edited the results section to clarify that FAA values were positive but non-significant compared to the other conditions:

“Posthoc testing following a main effect of condition indicated positive but non-significant FAA values (i.e., neutral neurobiological state) for both the No Connection condition (EMM = .120, CI = -0.06-0.3) and the Embrace condition (EMM = .116, CI = -0.06 - 0.3).”

Reviewer 2 Report

Comments and Suggestions for Authors

The article is aimed at studying the peculiarities of synchronization of brain rhythms during non-verbal interaction between romantic partners.

The authors wanted to examine the link between nonverbal communication and neural synchrony among romantic partners.

The novelty of the study consists in an attempt to examine the synchronization of neurobiological states over time during dyadic nonverbal connection.

The conclusions are consistent with the evidence and arguments presented.

The references are appropriate.

I have a few comments about the article revisions:

In the first block of the article, there is an error in the section numbering. There is 1.1, but no 1.2. Please revise.

In Discussion, you may consider synchronization results from other physiological signals (e.g. heart rate).

Please add a clear statement of the purpose of the hypotheses at the end of the introductory section. Right now these crucial elements are not explicitly present in the article.

Author Response

Thank you very much for taking the time to review our manuscript. We believe the changes we have made have significantly improved the manuscript. Please find our response to your comments (bold) below in italics. We also have highlighted major changes in the manuscript and have tracked-changes for minor word and grammatical changes.

The article is aimed at studying the peculiarities of synchronization of brain rhythms during non-verbal interaction between romantic partners. The authors wanted to examine the link between nonverbal communication and neural synchrony among romantic partners. The novelty of the study consists in an attempt to examine the synchronization of neurobiological states over time during dyadic nonverbal connection. The conclusions are consistent with the evidence and arguments presented. The references are appropriate. I have a few comments about the article revisions:

Comment 1: In the first block of the article, there is an error in the section numbering. There is 1.1, but no 1.2. Please revise.

Response 1: Thank you for pointing this out to us. We have realized that in transferring our manuscript over to the journal specific template, we missed this section and have now added it back in. 

Comment 2: In Discussion, you may consider synchronization results from other physiological signals (e.g. heart rate).

Response 2: While examining synchrony of other physiological signals was outside the scope of this paper, we agree that it is an important consideration for future research. As such, we have added a discussion of this in section 4.4.

“Future research may also consider incorporating other physiological measures to better understand the relationship between biological synchrony and relationship outcomes. For instance, research has demonstrated that synchronous electrodermal activity predicts mutual romantic and sexual interest [60], and sexual satisfaction moderates heart rate synchrony [61]. While understanding causal interactions and how neurobiological synchrony aligns with the synchronization of other physiological rhythms among romantic partners engaged in nonverbal connection was outside the scope of the current study, the presence of lagged synchrony in our results demonstrates a need to better understand this in future research.”

Comment 3: Please add a clear statement of the purpose of the hypotheses at the end of the introductory section. Right now, these crucial elements are not explicitly present in the article.

Response 3: It was our intention to have this section in the first submission but made a clerical error when converting our manuscript to the journal’s template. We now explicitly state our hypotheses in section 1.2 in context of our study objectives.

Reviewer 3 Report

Comments and Suggestions for Authors

The paper has some innovations, clear logical thinking and perfect structure, but it needs some minor adjustments: (1)EEG acquisition is sensitive, and body movements, blinking, especially embrace will bring additional signal acquisition interference. Please describe in more detail how to overcome these interferences in different situations in the experiment. What are the targeted pretreatment measures? (2) It is suggested to increase the difference analysis of the experimental results of different types of participants, which will help to enhance the universality of the conclusions.

Author Response

Thank you very much for taking the time to review our manuscript. We believe the changes we have made have significantly improved the manuscript. Please find our response to your comments (bold) below in italics. We also have highlighted major changes in the manuscript and have tracked-changes for minor word and grammatical changes. 

The paper has some innovations, clear logical thinking and perfect structure, but it needs some minor adjustments:

Comment 1: EEG acquisition is sensitive, and body movements, blinking, especially embrace will bring additional signal acquisition interference. Please describe in more detail how to overcome these interferences in different situations in the experiment. What are the targeted pretreatment measures?

Response 1: We agree that EEG acquisition is sensitive to body movements and blinking that are inherent in embracing. While we tried to control for this during the experiment by using strategies to remind participants to remain as still as possible, the Embrace condition only asked participants to move to embrace each other once. Additionally, we used standard preprocessing steps to ensure that the data was as clean as possible before analyzing. To better describe this in context with our results, we have added the following to section 4.4:

“Additionally, EEG data is particularly sensitive to body and head movements that occur naturally when people embrace. While we attempted to mitigate this noise during the experiment (i.e., asking participants to hold the same embrace the entire time, reminding participants to move as little as possible) and in preprocessing of the data (e.g., artifact detection/rejection described in section 2.5), it is possible that noisy data may have influenced our results.”

Comment 2: It is suggested to increase the difference analysis of the experimental results of different types of participants, which will help to enhance the universality of the conclusions.

Response 2: Thank you for this suggestion. We agree that future studies should aim to recruit more diverse samples to better understand the generalizability of our conclusions. As such we have added the following to section 4.4.

“Similarly, individual differences at the neural level (e.g., skull thickness, baseline alpha power) and demographic level (e.g., race, relationship type) may influence FAA scores [59].  As such investigating these constructs in a more diverse sample of romantic partners (e.g., non-white, Hispanic/Latine) and romantic partnership types (e.g. non-monogamous, casually dating) while correcting for overall alpha power within an individual will strengthen the generalizability of these conclusions.”

Round 2

Reviewer 2 Report

Comments and Suggestions for Authors

Dear authors, thanks for considering my comments.